# Current ARTs, Virologic Failure, and Implications for AIDS Management: A Systematic Review

**DOI:** 10.3390/v15081732

**Published:** 2023-08-13

**Authors:** Frank Eric Tatsing Foka, Hazel Tumelo Mufhandu

**Affiliations:** Department of Microbiology, Virology Laboratory, School of Biological Sciences, Faculty of Natural and Agricultural Sciences, North West University, Mafikeng, Private Bag, Mmabatho X2046, South Africa

**Keywords:** HIV, AIDS, antiretroviral therapy, drug resistance, virologic failure

## Abstract

Antiretroviral therapies (ARTs) have revolutionized the management of human immunodeficiency virus (HIV) infection, significantly improved patient outcomes, and reduced the mortality rate and incidence of acquired immunodeficiency syndrome (AIDS). However, despite the remarkable efficacy of ART, virologic failure remains a challenge in the long-term management of HIV-infected individuals. Virologic failure refers to the persistent detectable viral load in patients receiving ART, indicating an incomplete suppression of HIV replication. It can occur due to various factors, including poor medication adherence, drug resistance, suboptimal drug concentrations, drug interactions, and viral factors such as the emergence of drug-resistant strains. In recent years, extensive efforts have been made to understand and address virologic failure in order to optimize treatment outcomes. Strategies to prevent and manage virologic failure include improving treatment adherence through patient education, counselling, and supportive interventions. In addition, the regular monitoring of viral load and resistance testing enables the early detection of treatment failure and facilitates timely adjustments in ART regimens. Thus, the development of novel antiretroviral agents with improved potency, tolerability, and resistance profiles offers new options for patients experiencing virologic failure. However, new treatment options would also face virologic failure if not managed appropriately. A solution to virologic failure requires a comprehensive approach that combines individualized patient care, robust monitoring, and access to a range of antiretroviral drugs.

## 1. Introduction

Human immunodeficiency virus (HIV) incidence refers to the number of new HIV infections that occur in a particular population during a specific time period, usually expressed as a rate per 1000 or 100,000 people. The incidence of HIV varies widely between different populations and geographic regions and can be affected by a variety of factors such as access to healthcare, education, prevention programs, and cultural attitudes towards sex and sexuality [1]. As indicated in the recent data released by the Joint United Nations Programme on HIV/AIDS (UNAIDS), there were an estimated 1.5 million new HIV infections worldwide in 2020, representing a 23% decline in new infections since 2010 [2]. However, the incidence of HIV remains high in many parts of the world such as Eastern Europe, Central Asia, Latin America, the Middle East, and North Africa, where there has been an increase in annual cases of HIV infections over the past decades [3].

The Millennium Development Goals (MDGs) were established in 2000 as a global framework for reducing poverty and improving health outcomes in low- and middle-income countries. One of the MDGs was to combat HIV-acquired immunodeficiency syndrome (AIDS), malaria, and other diseases, with a specific target of halting and reversing the spread of HIV/AIDS by 2015 [4]. The global response to the MDG HIV/AIDS target was significant, with more than US $500 billion in spending on prevention, care, and treatment for HIV/AIDS between 2000 and 2015. This funding supported a range of interventions, including the scale-up of antiretroviral therapy (ART) for individuals living with HIV, the prevention of mother-to-child transmission (PMTCT) programs, and targeted prevention efforts among key populations. While there were some challenges and limitations to the MDG framework, particularly in terms of its narrow focus on select health outcomes and failure to adequately address broader social determinants of health, the global response to the HIV/AIDS target was widely hindered by the COVID-19 pandemic and conflicts in various parts of the world [2]. Consequently, in 2021, the UNAIDS reported 4000 new HIV infections every day with 650,000 deaths related to AIDS despite effective HIV treatment and strategies to prevent, diagnose, and cure opportunistic infections, with increasing inequality in access to HIV treatment coverage, especially by vulnerable communities [4].

Antiretroviral therapy (ART) refers to medicines that are used to manage AIDS resulting from HIV [5]. In the case of AIDS, ART involves the use of a combination of antiretroviral drugs (ARVs) that target different stages of the virus’s life cycle [5,6]. It has been shown to be highly effective at reducing the risk of transmission. In addition to their clinical benefits, ARVs have also shown important social and economic benefits, including better life quality, a decrease in the stigmatization and discrimination associated with HIV, and increased productivity and participation in the workforce.

There has been tremendous scientific progress in the design and development of antiretroviral therapies (ARTs). Early ARTs were developed based on the observed low count of CD4^+^ T-Cells in clinical symptoms of AIDS as compared to the most recent ones that are administered immediately after a patient is diagnosed regardless of the level of CD4^+^ cells [7]. As indicated by the WHO, current ART regimens are most often a combination of at least two or three active medicines from two or more drug classes, and wide access to ARTs has led to the provision of well-tolerated, less toxic, and more effective regimens. However, specific combinations of drugs used for HIV treatment vary depending on factors such as the individual’s viral load, CD4 count, and potential drug interactions [7]. Regrettably, the widespread use of ARTs has led to the emergence of resistant HIV strains and virologic failures. Virologic failure refers to a situation where ART is unable to effectively suppress or maintain the replication of HIV (viral load) below 200 copies/mL [8]. Several factors can cause virologic failure, and these include drug resistance, drug toxicity, drug interaction, poor adherence to ART, and underlying medical conditions. When virologic failure occurs, it can lead to a number of negative outcomes such as increased viral load, risk of HIV-related illness and transmission, and the development of drug resistance [9,10]. Assessing and understanding the mechanisms and factors of virologic failure and drug resistance, concerning previous versions and current versions of ARTs, is of utmost importance in the development of future ARTs. Therefore, the main aim of this article is to review current ARTs and, most importantly, their mechanism of action and the mechanism of HIV resistance to these drugs. Moreover, the article aims to point out the implications of HIV resistance on the management of AIDS, and briefly discuss prospective ART drug candidates.

## 2. Methodology

Data on ARTs were gathered, synthesized, and analyzed from literature databases. There was no date restriction to the publications that were selected and these were analyzed thoroughly as per the PRISMA guidelines [11].

### 2.1. Research Enquiry and Question Formulation

As described previously, the “Problem, intervention, comparison, outcome, and study type (PICOS)” procedure was used to formulate the study question [12]. We considered clinical and preclinical trial studies, clinical reviews, computational investigations, experimental studies, and epidemiological surveys, as well as hypothetical and theoretical studies. Therefore, the research queries were structured in the following manner: What are the current antiretroviral drugs used for the management of HIV/AIDS? Are all the current ARTs effective against HIV/AIDS? Which are those that demonstrate low efficacy and what could be the cause of that observation? What are the implications and consequences of ARTs’ low efficacy on the management of HIV/AIDS?

### 2.2. Databases Used for Data Retrieval and Evaluation

The repositories that were explored included PubMed, Web of Science, Scopus, and Science Direct. The bibliography of pertinent documents containing any relevant study that could be interesting was assessed. The investigation terms were selected based on previously identified keywords such as antiretroviral therapy (ART), HIV, AIDS, drug resistance, and virologic failure.

### 2.3. Criteria Determining the Study Selection

Studies and reports written in languages other than English were not included in this review. Additionally, abstracts, studies, meetings, and theses that were not relevant to the research theme were also not included in this report. No time or date restriction was set.

### 2.4. Data Analysis and Possible Sources of Bias

Selected reports were reviewed by the authors to ensure their relevance to the scope of this study. Full-text manuscripts were obtained, and all sections, including the abstract, introduction, methodology, results, conclusion, references, tables, and figures, were examined for data collection in an independent manner. Our criteria for selection and the possibility of data omission were the only potential sources of bias.

## 3. Results

Our systematic investigation, as demonstrated in the PRISMA flow diagram (Figure 1), resulted in 1233 research papers: 546 in PubMed, 279 in Web of Science, 314 through Scopus, and 94 through Science Direct. Out of these, 1044 were excluded as they either contained redundant information or were irrelevant to the research topic. After an in-depth assessment of titles, abstracts, and full texts, 176 studies were selected that met the eligibility criteria for this manuscript.

## 4. Preliminary Analysis of the Data Retrieved from the Literature

As HIV/AIDS swept across the globe to become a major public health issue in the 1980s, several pharmaceutical companies invested considerably in the development of therapies that could slow down the progression of this deadly disease. Azidothymidine (3’-azido-3’-deoxythymidine, zidovudine, or AZT) was the first drug to be validated by the Food and Drug Administration (FDA) for the management of HIV infection in 1987 [13]. The rate of mortality resulting from AIDS was still high until novel therapeutic options involving drug combinations were developed [14]. Such combinations involved three or more medicines directed towards two or more sites. Highly active antiretroviral therapy (HAART) was introduced, and despite the fact that it was life-saving, the initial treatment regimens were far from ideal as the daily dosing was complicated and the side effects were onerous [15]. Frequent occurrences of drug toxicities, challenges in maintaining long-term adherence, and the emergence of drug resistance often resulted in virological failure and clinical progression of the disease. Subsequently, a significant therapeutic breakthrough was achieved with the approval of Atripla^®^, the first single-tablet regimen taken once daily, consisting of a combination of tenofovir disoproxil fumarate (TDF), emtricitabine, and efavirenz [16], which was found to be more effective and was then often used [17]. In this section, we focused on a few drugs to briefly review the current classes of medicines used in HIV/AIDS management, their mode of action, their metabolic processes, their side effects, and their mechanism of resistance wherever applicable.

### 4.1. Nucleoside/Nucleotide Reverse Transcriptase Inhibitors (NRTIs)

#### 4.1.1. NRTIs Mode of Action

NRTIs (such as 3’-azido-3’-deoxythymidine, zidovudine, or AZT) were the first types of ARVs to be developed and approved by the FDA [13,16]. NRTIs are a class of antiretroviral drugs used in the treatment of retroviral infections such as HIV through the inhibition of the activity of the reverse transcriptase enzyme, which is an essential enzyme for the replication of retroviruses (Table 1). The mechanism of action of NRTIs involves their structural similarity to natural nucleosides and nucleotides such as dATP, dCTP, dTTP, and dGTP. When taken up by the infected cells, the NRTIs are phosphorylated to their active triphosphate forms, which can then compete with natural nucleotides for incorporation into the viral DNA by reverse transcriptase (Figure 2) [18].

NRTIs consist of either a nucleoside or nucleotide as a base, and they lack a 3’-hydroxyl group at the 2’-deoxyribosyl moiety. This absence of the 3’-hydroxyl group hinders the formation of a 3’-5’-phosphodiester bond in developing DNA chains, allowing NRTIs to obstruct viral replication. In particular, the NRTIs that exhibit lipophilicity, such as tenofovir disoproxil fumarate (TDF) and tenofovir alafenamide (TAF), as well as AZT, abacavir (ABC), and stavudine (d4T), are capable of permeating cellular membranes via non-facilitated diffusion, largely attributed to their hydrophobic properties [18,40]. One intriguing aspect of these drugs is that they can impede the creation of positive or negative strands of DNA by incorporating themselves during RNA-dependent DNA synthesis [18].

NRTIs have a higher affinity for the viral reverse transcriptase enzyme than for the human DNA polymerase, which reduces their toxicity to the host cells [40].

#### 4.1.2. Metabolism and Side Effects of NRTIs

The degradation mechanisms of NRTIs differ and are influenced by the type of cell in which they are present. However, their degradation is carried out in the liver where hepatic enzymes that participate in the purine or pyrimidine nucleoside salvage pathway are primarily responsible for breaking them down [41]. Additionally, NRTIs that do not undergo systemic absorption are eliminated by the kidneys, either by diffusion or through carrier-mediated transport [42]. The ATP-binding cassette (ABC) superfamily transporters play a crucial role in the NRTI’s efflux process from the cells. P-glycoprotein transporters (Pgp), breast cancer resistance protein transporters (BCRP), and multidrug resistance-associated protein transporters (MRPs) are the principal members responsible for the efflux of NRTIs. Only a small quantity of NRTIs are excreted in their unaltered form through the urine [43]. Tenofovir (TFV) is primarily eliminated from the body, along with uric acid through faeces and/or urine. The renal elimination of certain NRTIs, including TDF, involves active tubular secretion into the proximal tubule prior to elimination, and this is one reason for TDF nephrotoxicity [42,44]. Furthermore, certain enzymes such as purine nucleoside phosphorylase (PNP) are involved in the hydrolysis or phosphorolysis of guanosine and inosine nucleosides, thereby eliminating other NRTI drugs such as didanosine (ddI) from the system [41]. Despite the fact that NRTIs as antiretroviral drugs have increased the life expectancy of HIV-1 infected individuals, they do present setbacks such as the requirement of high daily doses, their rate of catabolism and elimination, their toxicity, and the fact that they are not curative drugs which therefore require life-long intake [45]. Moreover, side effects such as kidney dysfunction, hepatic steatosis (liver dysfunction), lactic acidosis, pancreatitis, cardiomyopathy, myopathy, and peripheral neuropathy have been reported as a result of mitochondrial malfunctioning [44,45,46,47,48].

#### 4.1.3. Mechanism of Resistance to NRTIs

Most first- and second-line antiretroviral therapy regimens for HIV-1-positive individuals are based on nucleoside/nucleotide reverse transcriptase inhibitors (NRTIs) [40]. The long-term use of NRTIs has been shown to cause resistance to one or multiple clinically used NRTIs and the development of NRTI cross-resistance has led to the failure of ART [40]. Resistance to NRTIs occurs because of mutations in the reverse transcriptase (RT) genome, which arise through amino acid substitutions in the transcribed enzyme. These substitutions cause structural changes in the active site or functional sites of the enzyme, enabling it to function in the presence of the NRTIs [49,50]. It is advisable to consult resources such as the HIV Stanford Database regularly, as newer data on drug resistance attributes may be available [22]. However, selected mutations resulting from NRTI usage are highlighted in Table 1. The accumulation of these mutations is responsible for the occurrence of cross-resistance against different drugs within the NRTI class [21]. Currently, NRTIs still constitute the backbone of ART to effectively control the progression of HIV infection for an extended period. Enhancing patient adherence can prevent the development of drug resistance while maintaining the therapeutic potential of ART [5].

### 4.2. Non-Nucleoside Reverse Transcriptase Inhibitors (NNRTIs)

#### 4.2.1. NNRTIs Mode of Action

NNRTIs are commonly used in conjunction with other antiretroviral drugs to manage HIV-1 infection. NNRTIs are a class of hydrophobic compounds with varying structures [51]. The primary function of NNRTIs is to obstruct the replication of HIV-1 by hindering the completion of the reverse transcription of the single-stranded RNA genome into DNA by reverse transcriptase (RT) (Figure 2). They do not necessitate intracellular metabolism to be effective, thereby distinguishing them from NRTIs [52]. However, some NNRTIs, such as efavirenz, can impede the late stages of HIV-1 replication by interfering with the processing of the HIV-1 Gag-Pol polyprotein. Others, such as the pyrimidinediones, have the ability to inhibit both HIV-1 RT-mediated reverse transcription and the entry of HIV-1/HIV-2 [53]. FDA-approved NNRTIs are listed in Table 1. First-generation NNRTIs include nevirapine (NVP), delavirdine (DLV), and efavirenz (EFV) while etravirine (ETR) and rilpivirine (RPV) are second-generation NNRTIs [54].

#### 4.2.2. Metabolism and Side Effects of NNRTIs

In terms of pharmacokinetics, the absorption and distribution of NNRTIs are heavily influenced by drug transporter activity, particularly P-glycoprotein activity [55]. Upon administration, NNRTIs are rapidly absorbed from the gastrointestinal tract and bind extensively to plasma proteins. The metabolism of these compounds is carried out by cytochrome P450 enzymes (CYP), mostly by CYP3A4 and glucuronoconjugation, with varying amounts of the administered drug being released in the urine and faeces either as metabolites or unchanged [56,57]. The effects of NNRTIs on other medications can vary, with some acting as inducers and others as inhibitors of drugs metabolized by CYP. However, NNRTIs interact well with CCR5 antagonists and protease inhibitors as well as other drugs such as antidepressants, analgesics, antifungals, anticoagulants, and antiarrhythmics [18,20,21]. For example, NVP is highly hydrophobic and lipophilic with high intestinal and placenta permeability (which is why it is recommended for pregnant women) [58].

NVP is subjected to hepatic biotransformation via CYP to generate several hydroxylated metabolites (2-, 3-, 8- and 12-hydroxynevirapine) and it also auto-induces isoenzymes 3A4 (CYP3A4) and 2B6 (CYP2B6) [56]. Another example is delavirdine (DLV). As opposed to the other NNRTIs, which induce enzymatic activity, DLV is metabolized in the liver by CYP3A4 but exerts an inhibitory effect on CYP3A4, CYP2C9, CYP2C19, and CYP2D6 and is excreted unchanged in the urine [57]. Second-generation NNRTIs such as ETR are highly lipophilic and are excreted through faeces, bile, and urine. ETR metabolism is mainly achieved through several CYP isozymes (CYP3A, CYP2C9 and CYP2C19) and the predominant metabolic pathway involves methyl hydroxylation, followed by the glucuronidation of the resulting metabolites [54].

Although the second-generation NNRTIs are generally safe and well-tolerated, NVP use is associated with hepatotoxicity (leading to fulminant hepatitis), Stevens–Johnson syndrome, severe rash, and epidermal necrolysis [58]. EFV is associated with side effects on the central nervous system (CNS) (toxicity, headache, dizziness, insomnia, impaired concentration, agitation, amnesia, somnolence, abnormal dreams, fatigue, and hallucinations), teratogenic side effects, maculopapular eruptions, skin rashes, cholesterolaemia, gastrointestinal side effects (nausea, diarrhoea, vomiting, dyspepsia, anorexia and malabsorption), lipodystrophy, moderate to severe pain, abnormal vision, arthralgia, asthenia, dyspnoea, gynecomastia, myalgia, myopathy, tinnitus, erythema multiforme, and Stevens–Johnson syndrome (rarely) [58,59]. DLV is associated with adverse effects such as teratogenicity (hence not recommended for pregnant women), CNS adverse effects (anxiety, depressive symptoms, and insomnia), respiratory system side effects (bronchitis, pharyngitis, sinusitis, upper respiratory infection, and cough), digestive system adverse effects (nausea, vomiting, and diarrhoea), asthenia, headache, flulike syndrome, localized pain, fever, and generalized abdominal pain [54].

#### 4.2.3. Mechanism of Resistance to NNRTIs

NNRTIs of the first generation have a low genetic barrier, and a single mutation is sufficient to confer resistance, leading to cross-resistance among these drugs. However, second-generation NNRTIs display higher genetic barriers to resistance. Moreover, NNRTIs required twice-daily dosing, which was a challenge for patients to adhere to, until the availability of Atripla^®^. Therefore, second-line regimens typically do not include NNRTIs due to an elevated risk of treatment failure associated with resistance. HIV-1 strains with reduced susceptibility to NVP, ranging from 100- to 250-fold, have also been observed in cell culture [60]. Mutations in the HIV-1 RT gene, specifically Y181C and/or V106A, have been identified through genotypic analysis, with the mutations varying depending on the virus strain and cell line used. While the most frequent resistance-associated mutation observed in vivo is Y181C, other NPV substitution mutations have also been reported (Table 1) [19,60]. DLV-associated resistance occurs mostly through substitutions at codon positions 103 and/or 181. The main mechanism of EFV viral resistance or decreased susceptibility is mutations in the RT gene. EFV, like other NNRTIs, displays low genetic barriers to resistance and therefore causes mutations that primarily affect the regions of the RT gene. The K103N mutation is the most well-known EFV resistance-associated mutation together with the substitution mutations as listed in Table 1 [22,60]. Second-generation NNRTIs have an improved resistance profile compared to first-generation compounds (due to a higher genetic barrier to resistance). However, resistance to these compounds has also been reported [15,22].

### 4.3. Protease Inhibitors (PIs)

#### 4.3.1. PIs Mechanism of Action

Protease Inhibitors (PIs) mimic the substrates of the HIV aspartyl protease enzyme, which plays a crucial role in processing viral proteins. The PIs attach to the active site of the enzyme to disrupt the viral maturation process, leading to an inability to create functional virions (Figure 2) [61]. Typically, these drugs are utilized in the second line of HAART and work synergistically with reverse transcriptase inhibitors or integrase inhibitors [25]. Approved PIs are listed in Table 1 and a few include atazanavir (ATV), lopinavir (LPV), and darunavir (DRV) (Table 1).

#### 4.3.2. Metabolism and Side Effects of PIs

The liver’s cytochrome P450 (CYP) enzyme system, particularly the CYP3A4 isoform, is responsible for the metabolic breakdown of all presently available protease inhibitors, and once metabolized, they are excreted through the urine and faeces [25]. The most frequent adverse reactions related to HIV protease inhibitors are the metabolic syndromes induced by these drugs, which include dyslipidaemia, insulin resistance, and lipodystrophy/lipoatrophy [25,62,63,64,65]. PI-induced metabolic disturbances result from mitochondrial toxicity, oxidative stress that modifies the secretion of adipokines, the inhibition of adipocyte differentiation, the non-competitive inhibition of GLUT-2/4 transporters, and the activation of the inflammasome [25]. Furthermore, cardiovascular and cerebrovascular diseases are also prevalent side effects [62,63,64,65].

#### 4.3.3. Mechanism of Resistance to PIs

Protease inhibitors exhibit a high genetic resistance barrier by serving as competitive inhibitors that bind to the active site of HIV-1 protease. This typically blocks the enzyme from digesting the Gag and Gag/Pol polyprotein precursors essential for virus maturation [66]. Resistance mutations differ across various codons in different HIV-1 subtypes. Codons 33, 34, 58, 63, 73, 71, 77, and 84 are more commonly associated with resistance mutations in subtype B viruses compared to subtype F or C viruses, whereas mutations at codons 20, 36, and 89 are not very common [67]. In subtype C virions, resistance mutations occur with higher frequency at codons 20, 36, 89, and 93, while being less common at codons 10, 30, 43, 46, and 74 [67].

There are two types of PI-resistance mutations: major and accessory mutations. While both types decrease susceptibility to one or more PIs, accessory mutations require the presence of major mutations to be effective. Some major mutations such as D30N, V32I, M46IL, G48VM, I50VL, I54VTALM, L76V, V82ATFS, I84V, N88S, and L90M enhance high-level resistance to only one PI, while others reduce the susceptibility of more than two PIs. For example, this is seen with D30N and I50L, which affect atazanavir and nelfinavir, respectively [22,68].

### 4.4. Integrase Strand Transfer Inhibitors (INSTIs)

#### 4.4.1. INSTIs Mechanism of Action

HIV utilizes the integrase (IN) enzyme to integrate its viral DNA into the host cell’s DNA via two catalytic steps, namely 3’ processing and strand transfer. INSTIs inhibit the function of the integrase enzyme in the virus, which is responsible for integrating the HIV-1 genetic material into the host DNA (Figure 2). This inhibition occurs by preventing the proper placement of viral DNA at the enzyme’s active site and by binding to the catalytic divalent cations (Mg^2+^ or Mn^2+^), required for its enzymatic activity, which are found within the retroviral integrase’s active site [27]. There are currently four antiretroviral drugs of the INSTI class that are FDA-approved for HIV treatment, and these include raltegravir (RAL), elvitegravir (EVG), bictegravir (BIC), dolutegravir (DTG), and cabotegravir (CAB) [10].

#### 4.4.2. Metabolism and Side Effects of INSTIs

Unlike other INSTIs that undergo metabolism by both cytochrome P450 (CYP) 3A4 and UDP-glucuronosyltransferase (UGT) 1A1, RAL is primarily metabolized solely by UGT1A1 through hepatic glucuronidation. RAL does not act as an inhibitor, inducer, or substrate of CYP enzymes. On the other hand, EVG is predominantly metabolized by CYP3A through oxidative metabolism and secondarily by UGT1A1/3 through glucuronidation [69]. DTG is primarily metabolized by UGT1A1, with minor involvement from CYP3A enzymes. In vitro studies have shown that DTG is a substrate for UGT1A1/3 and UGT1A9 enzymes, as well as the drug transporters breast cancer resistance protein (BCRP) and P-glycoprotein (P-gp) [10,70]. Furthermore, DTG exhibits the inhibition of the organic cation transporter (OCT)2 and multidrug and toxin extrusion transporter (MATE)1 in vitro. BIC is metabolized by both CYP3A and UGT1A1 and like DTG, it exhibits the inhibition of OCT2 and MATE1 in vitro [69,70]. CAB is primarily metabolized by UGT1A1, with some involvement from UGT1A9. In vitro studies have shown that CAB inhibits organic anion transporters (OAT)1 and OAT3, potentially leading to an approximately 80% increase in the area under the curve (AUC) of drugs that are substrates of OAT1/OAT3. INSTIs are cleared from the body through the urine and faeces [27,69].

The most common side effects of INSTIs include weight gain, neuropsychiatric side effects, irritability, anxiety, depression, headaches, fever, mood changes, insomnia, dizziness, fatigue, muscle pain, nausea, diarrhoea, rashes, and jaundice [27].

#### 4.4.3. Mechanism of Resistance to INSTIs

The fundamental mechanism of resistance to integrase has been thoroughly described and it typically starts with an initial mutation that decreases the drug’s binding affinity while potentially compromising virus fitness. Subsequent exposure to selective drug pressure results in the emergence of secondary resistance substitutions, which ultimately enhance viral fitness. The identified mutations induce changes in the activities of the integrase protein [71].

Resistance to first-generation INSTIs, such as RAL, occurs when specific amino acid substitutions take place. These substitutions involve Tyr143 (Y143H/R/C), Gln148 (Q148/H/R/K), or N155H, along with one or more additional substitutions (L74M, E92Q, Q95K/R, T97A, E138A/K, G140A/S, V151I, G163R, H183P, Y226C/D/F/H, S230R, and D232N), which ultimately result in RAL resistance [22,27]. Reduced susceptibility to EVG is observed when primary integrase substitutions, T66A/I, E92G/Q, S147G, and Q148R occur. Among these, E92Q is the most frequently encountered initial mutation associated with the failure of EVG-based regimens, followed by N155H and Q148H/R/K. The presence of E92Q results in the development of high-level resistance to EVG and intermediate-level resistance to RAL [22,27,72]. For the second-generation INSTIs, amino acid substitutions E92Q, G118R, S153F/Y, G193E, and R263K are associated with a two- to four-fold decrease in DTG susceptibility [73]. When combined with R263K, M50I is frequently selected in vitro by DTG and BIC, contributing to reduced DTG susceptibility. In vitro studies have shown that the substitutions R263K, E92Q, Y143R, N155H, and Q148R confer reduced susceptibility to BIC [70].

Despite their structural enhancements, second-generation INSTIs are still subject to resistance due to HIV’s ability to configure its magnesium ion binding geometry within the active site for its survival [27]. In fact, HIV-positive individuals who exhibited virologic failure to second-generation INSTIs were found to have no specific mutations in the integrase gene. One of the two extensively assessed alternative pathways through which HIV-1 develops resistance to INSTIs involves alterations to the 3-prime polypurine tract (3’-PPT). The 3’-PPT serves as a primer during the reverse transcription process, facilitating HIV-1 RNA conversion into double-stranded DNA. Studies have pointed out that this process leads to the accumulation of unintegrated viral DNA, which may account for INST treatment failure in the absence of integrase resistance mutations [23,69]. Besides the integrase gene mutation-induced resistance, another mechanism through which resistance to INSTIs has been observed is via mutations in the HIV-1 envelope glycoprotein complex (*env*). These *env* mutations enhance the efficiency of transmission between cells, indicating the potential for HIV-1 to evade the inhibitory effects of all antiretroviral drugs (ARVs) through mutations in *env* [28,74].

### 4.5. Entry Inhibitors (CCR5 Receptor Antagonists and Fusion Inhibitors)

#### 4.5.1. Mode of Action of CCR5 Receptor Antagonists

The binding of a chemokine coreceptor, CCR5 or CXCR4, is essential for HIV to enter the target cell and initiate infection (Figure 2). By interfering with the interaction between the gp-120 glycoprotein and the CCR5 chemokine receptor, CCR5 co-receptor antagonists impede the fusion of HIV with the host cell [75]. Within a decade after the identification of the HIV-1 coreceptor, multiple CCR5 antagonists were produced and evaluated as potential antiretroviral agents through clinical trials. Of all the CCR5 antagonists that were developed and clinically tested, maraviroc (MVC) was the most promising with proven safety tolerability and efficacy as compared to other candidates such as aplaviroc (APL), vicriviroc (VCV), and cenicriviroc (TBR-652), which were abandoned as a result of demonstrated hepatotoxicity, carcinogenesis, and suboptimal virologic activity leading to virologic failure [31,76,77,78,79]. CCR5 antagonists encompass different types, with prominent examples being chemokine derivatives, non-peptide small molecule compounds, monoclonal antibodies, and peptide compounds.

#### 4.5.2. Metabolism and Side Effects of Maraviroc

Results from investigations conducted on Caco-2 cells and P-glycoprotein (Pgp) homozygous null mice indicate the potential involvement of Pgp in the transmembrane transport of maraviroc, which may impact its absorption in the intestine as well as biliary excretion [32]. The data further reveal that maraviroc metabolism is primarily mediated by cytochrome P450 3A4 and is carried out in the liver [80]. The primary metabolite found in plasma is UK-408,027, which is a secondary amine resulting from N-dealkylation. Maraviroc also undergoes metabolism by CYP3A5, however, with CYP3A4 being the primary enzyme involved. CYP3A5 contributes to the formation of mono-oxygenated metabolites. Maraviroc demonstrates substantial distribution into the saliva, intestines, rectum, genital, and cerebrospinal fluids and tissues, and it is metabolized to many inactive metabolites and excreted from the body through the urine and faeces [80]. Possible maraviroc side effects include upper respiratory tract infections (sinusitis, bronchitis), cough and associated symptoms, pyrexia, rash, musculoskeletal or connective tissue symptoms, abdominal pain, constipation, appetite disorders, sleep disturbances, risks of hepatotoxicity, high lipoprotein cholesterol and triglyceride levels, postural hypotension, and myocardial ischemia or infarction [32,33].

#### 4.5.3. Mechanism of Resistance to CCR5 Receptor Antagonists

The precise mechanisms that underlie the development of resistance to maraviroc have not been thoroughly investigated. However, it is known that the resistance involves mutations that arise against the affinity of CCR5 bound to maraviroc on gp120. Resistant viruses are capable of interacting with CCR5, particularly through increased binding to the CCR5 N-terminal domain, despite the presence of maraviroc. Resistance to CCR5 receptor antagonists can occur either (1) as a consequence of the emergence and proliferation of X4-tropic strains of HIV-1 or through (2) mutations that confer resistance against maraviroc [31]. In the first case, resistance to Maraviroc commonly arises when X4-tropic HIV-1 is selected due to the pressure exerted by drug treatment. This occurs when R5-tropic viruses, which typically comprise the majority of viral strains, are effectively suppressed. As a result, the previously minor population of X4-tropic viruses multiplies and becomes the predominant viral species, leading to treatment failure [32,75]. There is evidence that naturally occurring mutations conferring resistance to maraviroc may be more prevalent in subtype-C HIV-1 compared to subtype-B. In fact, an analysis of 65 samples revealed that 52.3% of the samples (75% subtype C and 18.2% subtype B) exhibited at least one mutation associated with maraviroc resistance. Among these mutations, the A316T mutation in the gp120 region was frequently observed, occurring in 67.8% of subtype-C samples and 18.2% of subtype-B samples [34,35,81]. The most frequently observed mutations included G11R, P13R, and A25K.

As reported after in vitro studies, the genetic sequencing of subtypes-B and -G primary HIV-1 isolates following sequential passage revealed substitutions at specific positions. These included A316T, I323V, and A319S. Some viral isolates also exhibited deletions of isoleucine at position 315 and serine at position 317. Additionally, mutations were observed outside the V3 loop, across various regions including the V1, V2, V4, and constant domains (C)-3, C4, and C5 regions of gp120, as well as in gp41 [81]. V169M and N192K in the V2 region, L317W in the V3 region, I408A in the V4 region, D462N, N463T, S464T, and N465D in the V5 region, L820I, I829V, and Y837C mutations in gp41 were also observed [35]. Furthermore, mutations G11S + I26V, S18G + A22T, A19S + I26V, I20F + A25D + I26V, I20F + Y21I, P/T308H, T320H, and I322aV in the V3 loop were observed in patients demonstrating resistance to maraviroc [82].

### 4.6. Other HIV-1 Virus Entry Inhibitors

There are several advantages to focusing on the HIV-1 entry process as a target. Firstly, unlike protease inhibitors, targeting the entry process prevents the virus from integrating its genome into the host cell genome, thereby blocking the formation of latent viral reservoirs [36]. Secondly, entry inhibitors do not require entry into cells like reverse transcriptase, integrase, and protease inhibitors. Thirdly, the entry process consists of specific steps, each of which can be targeted for inhibition, providing multiple targets for entry inhibitors that are unlikely to develop cross-resistance [83,84]. Besides entry inhibitors that target CCR5 receptors discussed in the previous section, entry inhibitors that target the CD4 receptors [26,36,83], the hydrophobic pocket of gp41 [85,86,87], the MPER [88,89] and peptide-based entry inhibitors [90,91] have been studied and some have been developed and validated for clinical use by the United States Food and Drug Administration (FDA).

#### 4.6.1. Monoclonal-Antibody Based (mAb) Antiretroviral Therapy

Monoclonal antibodies (mAbs) form a major class of biotherapeutics that are considered antiviral bioagents because of their safety and specific virus-neutralizing ability [92]. The role of HIV-1 envelope glycoprotein in transmitting HIV virions into target cells makes it a major target for mAbs. Monoclonal antibodies target specific epitope regions of the viral envelope trimer (*Env*) to prevent viral entry through interference with the host receptors/co-receptors (CD4/CCR5) [93]. There are five essential trimeric *Env* regions that antibodies can bind to in order to elicit their neutralization function; that is, the CD4 binding sites (CD4bs), variable loop 2 (V2)-apex, V3-glycan, glycoprotein (gp)41/gp120 interface, and membrane-proximal external region (MPER) [86,87,94].

A wide array of mAbs have been investigated for their potential in the treatment of HIV/AIDS [39]. Ibalizumab, a recombinant humanized monoclonal antibody, was developed as the first intravenous HIV-1 treatment option and approved by FDA in 2018 for the management of multi-drug resistant (MDR) HIV-1 infection. It is administered alongside other forms of antiretroviral therapy and is specifically designed for treatment-experienced adults who are not responding to an antiretroviral regimen. It is a mAb-CD4-specific antibody that targets an epitope near the D1-D2 junction, which is on the opposite side of the gp120 binding site on CD4 [36,92,95]. Despite being an expensive treatment with the possibility of resistance due to the reduced expression or loss of N-linked glycosylation sites in V5 of gp120, ibalizumab remains highly significant for MDR patients [37].

CD4 interacts with gp120 through its residue Phe43, which occupies a conserved cavity formed by both the inner and outer domains of gp120 [96]. To target the CD4 binding site, several CD4-mimetic miniproteins have been developed [97]. The most successful construct, M48U12, effectively mimics CD4 by binding to the Phe43 cavity of gp120. It exhibits the potent inhibition of viral infection in the low nanomolar range [86]. Additionally, these miniproteins show resistance to acidic pH and high temperatures, indicating their potential as drug candidates [86,97]. An advantage of mAbs over other ARTs is their potential to disrupt both viral latency and persistent infections and their non-toxicity effect which arises with other ARTs’ administration. However, their disadvantages are that curative mAbs are yet to be developed, the development/production of mAbs is time-consuming, the production method has a relatively low throughput, and the produced mAbs cannot recognize new neutralizing epitopes of the virus [98]. Additionally, mAbs are produced via cell line immortalization by Epstein–Barr virus (EBV) and this constitutes a potential source of cancerous agent transfer [97].

#### 4.6.2. Peptide-Based Inhibitors (Fusion Inhibitors)

The formation of the six-helix bundle structure of gp41 plays a crucial role in viral infectivity. Consequently, numerous peptide-based inhibitors have been developed to block this step. Peptides derived from the heptad repeat domains (HR1 or HR2) were synthesized to bind to the fusion intermediates of gp41 and impede the formation of the six-helix bundle [92]. Notably, T20/enfuvirtide, a 36-residue peptide derived from gp41, became the first fusion inhibitor approved by the USFDA [99]. However, enfuvirtide has certain limitations for long-term use, such as the requirement for low-temperature storage, fresh reconstitution, and twice-daily subcutaneous injections. Additionally, injection site reactions, the rapid emergence of drug-resistant viruses, and high production costs have hindered its widespread adoption [38]. Similar peptide fusion inhibitors such as T1144 and sifuvirtide have faced the same drawbacks [100,101]. However, albuvirtide, also known as FB006M, a fusion inhibitor derived from an HR-2 peptide, obtained marketing support in China for the management of HIV-1 infection [102]. By conjugating maleimidopropionic acid to the HR-2 peptide, it can bind to human serum albumin, effectively preventing protease degradation in vivo. This modification results in a half-life ten times longer than that of enfuvirtide [103]. N-phenyl-N-piperidin-4-yl-oxalamide (NBD) analogues of CD4-mimetic inhibitors such as NBD-556 demonstrated the ability to bind to the Phe43 cavity of gp120 and inhibit both virus–cell fusions at small concentrations [104]. A similar observation was made with an optimized derivative called NBD-11021 [105]. Furthermore, using a cell-based assay, Bristol–Myers Squibb developed a small-molecule inhibitor called BMS-806 (also known as BMS-378806) in 2003 [106]. Subsequently, optimized derivatives like BMS-488043, BMS-626529, and BMS-663068 exhibited greater potency [94,107]. BMS-663068, known as fostemavir, has gained FDA approval as the first attachment inhibitor for use in combination with other antiretrovirals. It is primarily intended for treatment-experienced patients with multidrug-resistant HIV-1 infection [108]. Additionally, the membrane-proximal external region (MPER) represents one of the most highly conserved regions within gp41 and is targeted by several extensively studied broadly neutralizing antibodies, including 2F5, 4E10, Z13e1, 10E8, and DH511 [98,109,110]. These antibodies exhibit virus-neutralizing properties by binding to the prehairpin intermediate state of gp41, aided by their lipid-binding activity. However, when administered subcutaneously, fusion inhibitors produce the following undesirable effects: pain and discomfort, itching, redness, diarrhoea, nausea, fatigue, insomnia, depression, weight loss, cough, muscle ache, and pneumonia [36].

Enfuvirtide resistance is associated with mutations found specifically in the binding site of gp41, spanning codons 36–45. When a single mutation occurs in the enfuvirtide site, susceptibility is reduced 10 times, while two mutations result in an efficacy reduction of approximately 100 times. The most commonly observed mutations related to enfuvirtide are G36D/E/V, V38E/A, Q40H, N42T, and N43D [111]. A study investigated the pathways and mechanisms of resistance to two designed short-peptide-based HIV-1 fusion inhibitors (MTSC22 and HP23) using escape HIV-1 mutants generated against SC22EK, a template peptide for these inhibitors. Two substitutions, E49K and N126K, situated at the N- and C-heptad repeat regions of gp41, respectively, were found to confer high resistance to the inhibitors and also resulted in cross-resistance to enfuvirtide (T20) and sifuvirtide (SFT). The resistance mechanisms induced by SC22EK involved several key factors: (i) a significant reduction in the binding affinity of the inhibitors, (ii) a pronounced enhancement in the interaction of the viral six-helix bundle, and (iii) a severe impairment of the functionality of the viral *Env* complex [112]. Furthermore, a recent study identified various polymorphisms in the envelope gp41 region [113]. The primary polymorphisms observed were R46K/M/Q, E137K, and S138A, while R46K/M/Q and S138A were predominantly found in subtype CRF07_BC, and E137K was prevalent in subtype B.

#### 4.6.3. Mechanism of Resistance to Ibalizumab

Resistance to ibalizumab primarily occurs through the reduction or loss of certain N-linked glycosylation sites within the V5 loop, along with specific alterations in the positions of these sites [114]. Resistance to ibalizumab can develop rapidly within 1 to 2 weeks when used as monotherapy. In some cases, a missed infusion of ibalizumab every 4 weeks resulted in decreased virologic response and raised concerns about the emergence of resistance [96,115]. Additionally, in vitro studies have shown increased infectivity potential in the presence of ibalizumab resistance [116]. There have been no reports of cross-resistance between ibalizumab and other antiretroviral treatments and in vitro studies have shown synergistic antiretroviral activity when ibalizumab was co-administered with enfuvirtide [117]. No studies have specifically assessed the efficacy of ibalizumab in HIV-1 types N or O infections or in different subtypes of HIV-1 type M infections. Genetic variability exists across various HIV-1 subtypes, with some clades showing a higher tendency for specific mutations [67]. Variations in the variable loops have been observed among different subtypes, and alterations can occur as HIV-1/AIDS progresses from acute to chronic infection [118].

## 5. Drivers of Virologic Failure

The recently introduced Global AIDS Strategy (2021–2026) aims to diminish the disparities that fuel the AIDS epidemic and prioritize individuals to ensure that the world is progressing towards eradicating AIDS as a public health risk by 2030 [2]. Although this strategy lays out an efficient framework of transformative actions to achieve this goal, virologic failure remains a serious source of concern that can dampen these efforts. Virologic failure is associated with high health costs and increased mortality rates due to the progression of AIDS. A significant proportion of people living with HIV who have access to healthcare services and ARTs still present unsuppressed virologic loads [9]. While virologic failure can be a complex and significant complication of HIV infection, advancements in diagnostics and innovative therapies have expanded the range of treatment options even for patients who have been extensively exposed to antiretroviral therapy (ART) and have developed multidrug-resistant strains of the virus. The next section sheds some light on the drivers of virologic failure and how it can be addressed.

### 5.1. Addressing Virologic Failure Resulting from Resistant Strains

In cases of virologic failure or concerns about incomplete virologic response, it is recommended to conduct genotypic resistance testing. The available literature indicates that resistance testing is valuable in guiding the selection of initial therapy and determining the optimal antiretroviral therapy (ART) regimen after treatment failure [119]. Numerous studies have shown the cost-effectiveness of genotypic resistance testing both before initiating ART and during instances of virologic failure [120,121,122,123,124]. However, it should be noted that pre-treatment integrase gene sequencing, particularly for individuals taking integrase strand transfer inhibitors (INSTIs), may increase costs and potentially lead to unfavourable clinical outcomes [125]. This is because the test results might discourage the use of effective regimens like dolutegravir (DTG) or bictegravir (BIC) [125]. Genotypic resistance testing is the preferred method to investigate resistance mutations in the HIV-1 genome. Sanger sequencing focuses on the reverse transcriptase (RT) and protease (PR) regions of the pol gene with a turnaround time of 7–14 days. Sequences of the integrase protein (IN) region and the env gene are not standardly assessed but should be requested with HIV viral tropism assays [126]. Phenotypic assays measure drug activity by culturing the virus in the presence of antiviral compounds, but they are less common due to higher costs and longer turnaround times. They are recommended for new or investigational agents or complex resistance cases [127]. Next-generation sequencing (NGS) is an advanced method used for genotypic resistance testing, utilizing high-throughput procedures, which require fewer expert workers and lesser costs per sample than Sanger sequencing [127]. NGS can detect minor variants at very low thresholds, as low as 1%, enabling the identification of a higher number of drug resistance mutations compared to traditional Sanger sequencing. However, a crucial challenge in this field is determining the ideal threshold for detecting mutations with NGS that correlates with clinically significant resistance [126]. Another challenge lies in the cost (about US $380 per test), which constitutes a significant barrier for HIV genotyping in resource-limited settings (RLS), and this can hinder the effective therapeutic management of HIV/AIDS patients in such settings [128]. Collaborative partnerships were initiated in some resource-limited countries to develop discounted in-house HIVDR genotyping assays that could be commercially affordable for such settings or similar countries [129,130,131].

For optimal clinical relevance, resistance testing should ideally be conducted while individuals are receiving antiretroviral therapy (ART) or within four weeks of discontinuing treatment [127]. This timeframe ensures that the testing captures potential resistance mutations in the HIV-1 strains. If a person has been off ART for an extended period, there is a possibility that the viral strain has reverted to wild-type, while resistant strains may exist in lower quantities, making them undetectable [132]. However, it is important to note that even in individuals who have ceased ART, significant mutations, particularly those related to non-nucleoside reverse transcriptase inhibitors (NNRTIs), can still be detected [9]. When selecting a new antiretroviral therapy (ART) regimen, it is crucial to take into account both current and previous resistance test results. If the selective pressure from a previous drug is no longer exerted, resistance to that specific drug may not be evident in the current HIV genotype. Nevertheless, if a mutation was identified in a previous genotypic resistance test, it should still be considered in the overall assessment of the genotype to account for cumulative resistance information.

Healthcare providers should select medicines from a new drug class or from a class previously used by the individual, provided that there is no evidence of cross-resistance based on resistance test results. Ideally, the regimen should include a minimum of two active agents from different drug classes, although three agents are preferred in some cases. Guidelines caution against adding just one active agent to a failing regimen due to the risk of treatment failure with functional monotherapy [127]. Furthermore, a second-line ART regimen can be considered when first-line regimens fail, but in low-income countries, about 18.8% of individuals living with HIV have encountered the failure of second-line treatment, and these drugs are unaffordable and not widely available for patients in such settings [133,134]. However, guidelines for the administration of second-line regimens in cases of first-line regimen failure have been outlined by the UNAIDS [2]. Salvage regimens or rescue therapy are indicated for ART-experienced individuals with few treatment possibilities. The occurrence of virologic failure in second-line and salvage regimens is challenging due to the significant ART exposure and the presence of resistance [135,136]. In such cases, the combination of a fusion inhibitor (enfurvitide), a mAb (ibalizumab), or a CCR5 antagonist (maraviroc) with an optimized regimen can be effective [137,138,139]. Additionally, specific regimens for the management of virologic failures in cases of existing co-infections such as tuberculosis, hepatitis B, and hepatitis C have been clearly outlined [127,140,141,142].

### 5.2. Addressing Virologic Failure Resulting from Poor Adherence to ART Regimens

In certain cases of virologic failure, some individuals may exhibit detectable viremia despite the absence of resistance mutations as identified by standard genotype testing. Virologic failure without resistance typically arises from insufficient drug levels, primarily due to non-adherence to the prescribed regimen. Other potential factors that can contribute to this situation include compromised gastrointestinal absorption and interactions between different medications [9]. Adherence to ARTs poses a multifaceted challenge and is influenced by various factors. Societal elements, such as drug abuse, financial problems, and concerns related to the stigmatization and non-divulgence of HIV status, should all be taken into account [143,144,145,146]. The presence of medical comorbidities, including coexisting mental disorders, may also heighten the risk of non-adherence to ART [147,148]. Additionally, aspects of the treatment regimen itself can hinder adherence, particularly if the number of pills to take is high or if the medication is poorly tolerated due to its side effects [148]. Lastly, medication stock-outs are common in resource-limited settings and this is another factor of non-adherence [2]. Furthermore, drug interactions and mistakes either in druggist’s dispensing or patient misinterpretation of how to take ARTs appropriately constitute another factor of poor adherence to ARTs. In this case, ensuring the accuracy and regularity of drug dispensing and pharmacy refill records could be helpful tools. Online tools can be used for assessing potential drug–drug interactions [149]. Moreover, it is important to note that certain antiretroviral medications, such as atazanavir, darunavir, and rilpivirine, should be taken with food to ensure ideal drug concentrations [150,151]. Additionally, medical conditions or anatomical disorders affecting absorption may result in reduced drug levels, thus healthcare providers should thoroughly evaluate patient symptoms and review their medical and surgical history when assessing cases of virologic failure [152].

Even though ART regimens have been simplified to one-pill doses nowadays, patient adherence to ARTs and assurance of therapeutic drug levels can further reduce treatment failure if patient-centred strategies to target barriers to adherence are implemented, drug substitutes with lesser side effects are provided, and pill text-reminders are given to patients [127,153]. Furthermore, counselling and psychological assistance for patients exposed to factors that can contribute to poor drug adherence can also contribute to reducing virologic failure [148].

## 6. Future Perspectives

Looking ahead, there are several promising perspectives in HIV drug design that hold the potential for advancing the treatment and management of the virus. One key area of focus is the development of novel antiretroviral agents with improved efficacy, safety, and tolerability profiles. Researchers are actively exploring new drug targets and mechanisms of action to overcome existing resistance and enhance viral suppression. Another area of interest is the exploration of long-acting formulations, such as injectables or implants, that offer the extended release of antiretroviral drugs. This approach aims to simplify treatment regimens by reducing the frequency of drug administration and improving medication adherence. Furthermore, the field of HIV drug design is embracing the concept of combination therapies that target multiple stages of the viral life cycle. By simultaneously attacking different vulnerable points in the HIV replication process, combination therapies have the potential to increase treatment efficacy, prevent drug resistance, and prolong viral suppression. In addition to traditional small-molecule drugs, there is growing interest in the development of biologic agents, such as monoclonal antibodies, which can specifically target HIV and enhance the immune response against the virus. These biologics hold promise for both treatment and prevention strategies.

One drug candidate is Fostemsavir (FTR), an attachment inhibitor, which has been approved by the FDA for the treatment of heavily treated adults with multidrug-resistant HIV-1 [154]. By inhibiting the conformational changes in HIV-1 gp120 necessary for CD4 attachment, FTR prevents viral entry into susceptible cells [155]. The efficacy and safety of fostemsavir were evaluated in a clinical trial called BRIGHTE, which included highly treatment-experienced patients who had failed their current regimen [156]. Over the 96-week trial period, fostemsavir consistently achieved virologic suppression, with increasing proportions of participants achieving an undetectable HIV viral load at week 24, week 48, and week 96. Additionally, fostemsavir showed CD4+ T-cell recovery, even in patients with severe immunosuppression at baseline [157]. Furthermore, a promising mAb candidate which acts as a CCR5 antagonist, Leronlimab (Pro140), is currently under investigation and has demonstrated antiviral effects in phase III clinical trials on SIV and HIV [158,159]. UB-421, another mAb candidate that acts as an attachment inhibitor, was proven to be effective against HIV strains that were resistant to broadly neutralizing Abs, entry inhibitors, and other ARTs [160]. Clinical studies have proven that “naturally occurring new generation broadly neutralizing anti-HIV-1 antibodies” (bNAbs) are harmless, have half-lives of about 2 to 3 weeks, and decrease viremia by 1.5 log_10_ copies/mL [161]. However, the selection of resistant strains was observed with bNAb monotherapy, but viral suppression was effectively controlled and the selection of new resistance traits was absent in patients who were treated with a combination of two bNAbs candidates, specifically 3BNC117 and 10–1074 [162]. Unlike ART, bNAbs can involve the host immune system by initiating Fc effector functions, which may result in the destruction of latently infected cells [163]. Even though significant effects of bNAbs on the latent HIV reservoir have not yet been proven, the potency and the extended half-lives of bNAbs combinations and/or multi-specific antibodies make them promising therapeutic options against HIV-1 [164].

A new category of ARTs, referred to as nucleoside reverse transcriptase translocation inhibitors (NRTTIs), is currently being investigated. Islatravir (ISL, MK-8591), a drug candidate belonging to this class, exerts its antiviral activity by inhibiting reverse transcriptase (RT) through various mechanisms such as RT translocation inhibition and delayed chain termination via viral DNA structural modifications [165]. Despite interesting results on multidrug-resistant HIV-1 strains and HIV-2, clinical investigations were halted due to an unexpected decline in total T-lymphocytes and CD4+ cell count [166]. Furthermore, an HIV-1 capsid inhibitor called Lenacapavir was approved in the USA and the European Union in 2022. It was proven to be effective against patients with limited treatment options and no resistance has been reported so far [167,168]. Long-acting HIV maturation inhibitors such as GSK2838232, GSK3640254, and GSK3739937 are also under investigation. They act by interfering with the protease-mediated cleavage of the HIV-1 gag protein precursor. This disruption results in the production of immature virions that are non-infectious [169,170,171]. Additionally, novel long-acting antiretroviral therapy (ART) formulations are currently being investigated to improve AIDS management. These are promising therapeutic options for virologically suppressed patients and those presenting multidrug resistance [172]. The first long-acting ART that was approved by FDA in the US is a dual-drug intragluteal injection referred to as long-acting cabotegravir/rilpivirine (CAB/RPV). This regimen is recommended every 4 weeks and is indicated for virally suppressed patients with no history of virologic failure [173]. The advantages of long-acting ART include less frequent dosing, fewer side effects, fewer drug–drug interactions, the inexistence of pill burden, high bioavailability, a reduced stigma associated with daily pill intake, and increased adherence [174]. However, long-acting ART setbacks include concerns for pregnant women, emerging resistance, visit inconsistencies of patients (the injection cannot presently be self-administered and the in-clinic aspect may constitute a problem for patients), the need to associate with an oral regimen for hepatitis B patients, and risks of virologic failure, especially in cases of non-adherence [175].

Lastly, advances in computational modelling, artificial intelligence, and machine learning are being harnessed to accelerate the discovery and optimization of new HIV drugs [176,177,178,179,180,181]. These technologies enable the rapid screening of large compound libraries, the prediction of drug–target interactions, and the design of novel drug candidates with improved properties. Furthermore, plant-based nanoparticles have shown potential as a novel approach in the fight against HIV. These nanoparticles, derived from various plant sources, can be engineered to carry antiretroviral drugs or therapeutic agents specifically targeting the virus [182,183,184]. By leveraging the unique properties of plant-based nanoparticles, such as their biocompatibility, biodegradability, and ability to encapsulate and deliver active ingredients, researchers aim to enhance the effectiveness and targeted delivery of HIV treatment [184,185]. Additionally, these plant-based nanoparticles offer the advantage of being derived from sustainable and renewable sources, making them an attractive option for developing eco-friendly and cost-effective HIV therapies [186,187]. While this area of research is still in its early stages, the use of plant-based nanoparticles also holds promise in advancing HIV treatment strategies.

## 7. Conclusions

Despite the availability of effective ARTs and the implementation of global initiatives to tackle HIV/AIDS progression, it still remains a global public health issue. This has been exacerbated by the constant emergence of resistant viral strains resulting from poor adherence to ART regimens, which consequently lead to virologic failure. Although the investigation of potential alternative treatment options could significantly ameliorate the clinical management of HIV/AIDS, this would still constitute a failure if specific factors of virologic failure are not initially addressed. Virologic failure can arise from many factors, such as inadequate treatment adherence, the development of drug resistance, suboptimal drug concentrations, drug interactions, and viral factors such as the emergence of drug-resistant strains. Virologic failure could be addressed if the following key aspects are taken into consideration: (1) conducting genotypic resistance testing; (2) differentiating between failure caused by poor adherence and failure caused by drug resistance, which can coexist; and (3) selecting optimized treatment regimens consisting of at least two active drugs from separate classes. Similar to the goals of treatment initiation, the objective of therapy following treatment failure would be to choose a regimen that is well-tolerated, affordable, minimally burdensome, and capable of rapidly and consistently achieving virologic suppression. The future of HIV drug design is focused on developing more effective, convenient, and personalized therapies that can achieve sustained viral suppression, improve patient outcomes, and ultimately move closer to the goal of eradicating HIV/AIDS.

## Figures and Tables

**Figure 1 viruses-15-01732-f001:**
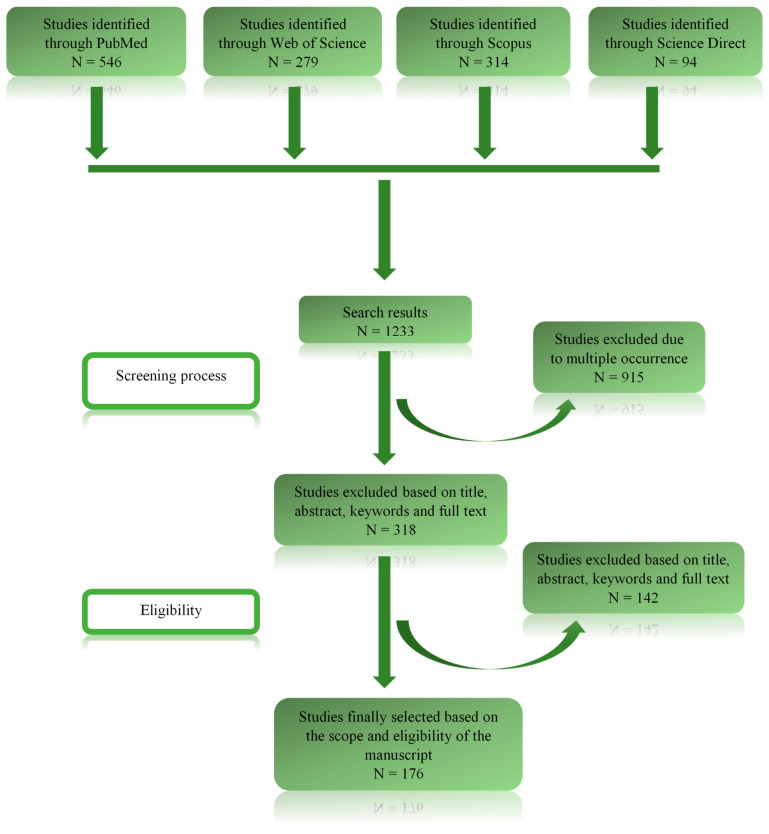
PRISMA flow diagram of this study.

**Figure 2 viruses-15-01732-f002:**
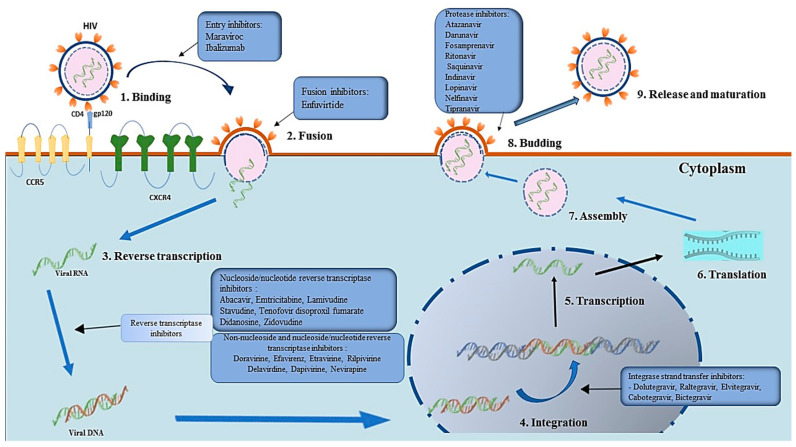
Mechanism of action of antiretroviral therapies.

**Table 1 viruses-15-01732-t001:** FDA-approved ART drugs and some of their reported mutations.

Class of ART		Drug	Commercial Name	Company	Mutations	References
Nucleoside/nucleotide reverse transcriptase inhibitors (NRTIs)		Abacavir (ABC)	Ziagen^®^	GlaxoSmithKline, Brentford, Middlesex, UK.	K65R, L74V, Y115F, M184V/I, M41L, D67N, T215Y, K219N/Q, A62V, K70R, V75I, F116Y, Q151M, L210W…etc.	[19,20,21,22]
		Emtricitabine (FTC)	Emtriva^®^	Gilead Sciences, Lakeside Drive Foster City, California, USA.	M184V/I, K65R, T69, Q151M, K43Q/N, E203K, H208Y, D218E, K223Q/E, L228H/R…etc.	
		Lamivudine (3TC)	Epivir^®^	GlaxoSmithKline, Brentford, Middlesex, UK.	M184V/I, E44D, V1181, M41L, D76N, M184V, T215Y, K219N, T215F, K70R, K65R, K219Q, L210W, A62V, L74V…etc.	
		Stavudine (d4T)	Zerit^®^	Bristol Myers-Squibb, Princeton Pike, USA.	M41L, M184V, T215Y, A62V, D67N, K70R, V75I, F116Y, Q151M. K219Q…etc.	
		Tenofovir disoproxil fumarate (TDF)	Viread^®^	Gilead Sciences Lakeside Drive Foster City, California, USA.	K65R, M41L, L210W, L74V, K70E/G/Q/T/N, S68G…etc.	
		Tenofovir alafenamide (TAF)	Vemlidy^®^	Gilead Sciences Lakeside Drive Foster City, California, USA.		
		Didanosine (ddI)	Videx EC^®^	Bristol Myers-Squibb, Princeton Pike, USA.	L74V, A62V, D67N, K70R, V75L, F116Y, Q151M, K219Q, M41L, M184V, L210W, T215Y, T215F…etc.	
		Zidovudine (AZT)	Retrovir^®^	GlaxoSmithKline, Brentford, Middlesex, UK.	K70R, T215T/F, M41L, K65R, D76N, M184V, T215Y, K219N, T215F…etc.	
Non-nucleoside and nucleoside/nucleotide reverse transcriptase inhibitors (NNRTIs)		Doravirine (DOR)	Pifeltro^®^	Merck & Co, New Jersey, USA.	A98G, L100I, K103N, V106A, V108I, Y188L, G190S, P225H, V106A, F227L, M230L, L234I, and Y318F	[22,23,24]
		Efavirenz (EFV)	Sustiva^®^	Mylan, Pennsylvania, USA.	K103N, N348I, Y318F, K238N/T, L234I, Y232H, F227L, P225H, G190A/E/Q, Y188L and Y181C	
		Etravirine (ETR)	Intelence^®^	Janssen, Beerse, Belgium.	V90I, A98G, L100I/V, K101E/H/P, V106A/I/M, L234I, E138A/G/K/Q, V179D/E/F/I/L/M/T, Y181C/I/S/V, Y188C/H/L, G190A/C/E/Q/S/T/V, P225H, F227C, M230L, and K238 N/T	
		Rilpivirine (RPV)	Edurant^®^	Tibotec, Mechelen, Belgium.	V90I, L100I, K101E/P/T, V106A/I, V108I, E138A/G/K/Q/R, V179F/I/L, Y181C/I/V, Y188I, G190E, H221Y, F227C/L, and M230I/L	
		Delavirdine (DLV)	Rescriptor^®^	ViiV Healthcare, Brentford Middlesex, UK.	P236L, K103N, Y181C and Y318F	
		Dapivirine (DPV)	-	Janssen Therapeutics, Beerse, Belgium.	L100I and K103N	
		Nevirapine (NVP)	Viramune^®^	Boehringer Ingelheim Pharmaceuticals, Ingelheim am Rhein, Germany.	Y181C, V106A, N348I, P236L, L234I, Y232H, G190A/E/Q, M230L, F227L, K103N, P225H, Y188L, Y181C/I/V…etc.	
Protease inhibitors (PIs)		Atazanavir (ATV)	Reyataz^®^	Bristol Myers Squibb Co. Princeton Pike, USA.	32I, 33F, 46IL, 47V, 48VM, 50L, 54VTALM, 82ATFS, 84V, 88S, 90M…etc.	[22,25,26]
		Darunavir (DRV)	Prezista^®^	AbbVie Inc, Chicago, USA.	32I, 33F, 47VA, 50V, 54LM, 76V, 82F, 84V…etc.	
		Fosamprenavir (FPV)	Lexiva^®^	GlaxoSmithKline plc, Brentford, Middlesex, UK.	32I, 33F, 46IL, 47VA, 50V, 54VTALM, 76V, 82ATFS, 84V, 90M…etc.	
		Ritonavir (RTV)	Norvir^®^	Abbott Laboratories, Illinois, USA.	10I, I54V, L63P, L76V, A71V, V82A/F, I84V K14R, K20I, E34Q, I47V, I54M, K55R, T74P and I84V	
		Saquinavir (SQV)	Invirase^®^	F. Hoffmann La Roche Ltd. Basel, Switzerland.	48VM, 54VTALM, 82AT, 84V, 88S, 90M…etc.	
		Indinavir (IDV)	Crixivan^®^	Merck & Co, Inc. New Jersey, USA.	32I, 46IL, 47V, 54VTALM, 76V, 82ATFS, 84V, 88S, 90M…etc.	
		Lopinavir (LPV)	Kaletra^®^	Abbott Laboratories, Illinois, USA.	32I, 33F, 46IL, 47VA, 48VM, 50V, 54VTALM, 76V, 82ATFS, 84V, 90M…etc.	
		Nelfinavir (NFV)	Viracept^®^	Agouron Pharmaceuticals, San Diego, California, USA. (Pfizer)	30N, 33F, 46IL, 47V, 48VM, 54VTALM, 82ATFS, 84V, 88DS, 90M…etc.	
		Tipranavir (TPV)	Aptivus^®^	Boehringer Ingelheim GmbH. Ingelheim am Rhein, Germany.	32I, 33F, 46IL, 47VA, 54VAM, 82TL, 84V…etc.	
Integrase strand transfer inhibitors (INSTIs)		Dolutegravir (DTG)	Tivicay^®^	ViiV Healthcare Brentford Middlesex, UK.	E92Q, N155H, G149A, S147G, G118R, S153F/Y, G193E, M50I, R263K, Q148H/K/R…etc.	[22,27,28,29,30]
		Raltegravir (RAL)	Isentress^®^	Merck & Co., Inc. New Jersey, USA.	E92Q, S153Y/F, Q148H, N155H, E157Q, Y143H/R/C, S147G, Q148/H/R/K, L74M, Q95K/R, T97A, E138A/K, G140A/S, V151I, G163R, H183P, Y226C/D/F/H, S230R, D232N…etc.	
		Elvitegravir (EVG)	Vitekta^®^	Gilead Sciences, Lakeside Drive Foster City, California, USA.	T66A/I, E92G/Q, S147G, R263K, Q148R, E157Q, N155H, S153Y/F, S147G, Q148H/K/R, V151I…etc.	
		Cabotegravir (CAB)	Vocabria^®^ and Apretude^®^	Janssen Pharmaceutical Companies (Beerse, Belgium) and ViiV Healthcare (Brentford Middlesex, UK)	H51Y, T66A/I/K, G149A, L74M/I/F, S153Y/F, N155H, G140S, Q148H, T97A, G118R, F121C, Q148H/K/R E138K/A/T…etc.	
		Bictegravir (BIC)	Biktarvy^®^	Gilead Sciences, Lakeside Drive Foster City, California, USA.	R263K, M50I, R263K, E92Q, S153Y/F, Y143R, N155H, G149A, Q148H/K/R, Q148R…etc.	
Entry inhibitors	ARVs that block viral entry (CCR5 receptors antagonists)	Maraviroc (MVC)	Selzentry^®^	Pfizer, New York, USA.	G11R, P13R, I408A, A316T, I323V, A319S, A25K, I315S, I317S, V169M, N192K, L317W, D462N, N463T, S464T, N465D, L820I, I829V, Y837C…etc.	[31,32,33,34,35]
	ARVs that inhibit HIV-1 virus fusion	Enfuvirtide (T-20)	Fuzeon^®^	Hoffmann-La Roche, Basel, Switzerland.	A30V, L33V, L34M, G36D/E/S/V, I37V, V38A/E, Q39H/R, Q40H, N42T/D, N43D, L44M, L45M, R46M, L54M, N140I, T18A, Q40H, L45M, T268A, N126K, E137K, S138A…etc	[26,36]
	Post-Attachment Inhibitors	Ibalizumab (IBA)	Trogarzo^®^	TaiMed Biologics Taipei, Taiwan.		[37,38,39]
Pharmacokinetic Enhancers		Cobicistat (COBI)	Tybost^®^	Gilead Sciences, Lakeside Drive Foster City, California, USA.	Not applicable	[15]

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
