# Peer review of "Current ARTs, Virologic Failure, and Implications for AIDS Management: A Systematic Review"

_viruses, 2023, doi:10.3390/v15081732_

Round 1
Reviewer 1 Report
Foka and Mufhandu present a thorough overview of virologic failure with current ART regimens and its implications for treatment. They thoughtfully incorporate the history of drug development and regimens, mechanisms and resistance in a succinct way that is enjoyable to read.
There were a few spots where clarification or corrections need to occur. I would highly encourage the authors to further comb through and make sure there are no other factual errors.
- Inappropriate use of HAART versus ART at line 143. While HAART is an older term, many of the regimens would still be considered HAART as they are 3 drug combinations.
- The term “fusion” of drugs at line 142 is confusing.
- Is figure 2 a reproduction from another paper?
- Line 184. TAF in inaccurately described as being nephrotoxic. TDF if the one which has nephrotoxicity concerns. Further the authors says that TFV is primarily eliminated as metabolites. Here is the text from the tenofovir package insert: “In vitro studies indicate that neither tenofovir disoproxil nor tenofovir are substrates of CYP enzymes. Following IV administration of tenofovir, approximately 70−80% of the dose is recovered in the urine as unchanged tenofovir within 72 hours of dosing.” Something woefully inaccurate calls into question other potential inaccuracies in subject areas I am less versed in.
- Line 215, why is the size of the molecule mentioned? Nearly all approved therapies are small molecules. If this is important it should be given some context.
There is a need to comment and add sections on
- long acting ART
- specifics of switching regimens for certain classes post-resistance development. Any clinical caveats to highlight?
- context of genotyping and impact on practice in low income settings
- mAb development and bnAb development
Authors and copy-editors should check for punctuation errors (e.g. line 659)
Author Response
Responses to reviewer 1:
Query 1: Inappropriate use of HAART versus ART at line 143. While HAART is an older term, many of the regimens would still be considered HAART as they are 3 drug combinations.
Answer to Query 1: This issue was addressed in the revised manuscript.
Query 2: The term “fusion” of drugs at line 142 is confusing.
Answer to Query 2: The term “fusion” was replaced by the term “combination”
Query 3: Is figure 2 a reproduction from another paper?
Answer to Query 3: No, figure 2 is not a reproduction from another paper.
Query 4: Line 184. TAF in inaccurately described as being nephrotoxic. TDF if the one which has nephrotoxicity concerns. Further the authors says that TFV is primarily eliminated as metabolites. Here is the text from the tenofovir package insert: “In vitro studies indicate that neither tenofovir disoproxil nor tenofovir are substrates of CYP enzymes. Following IV administration of tenofovir, approximately 70−80% of the dose is recovered in the urine as unchanged tenofovir within 72 hours of dosing.” Something woefully inaccurate calls into question other potential inaccuracies in subject areas I am less versed in.
Answer to Query 4: The sections containing these concerns were rewritten, taking into consideration the above information and other research outputs.
Query 5: Line 215, why is the size of the molecule mentioned? Nearly all approved therapies are small molecules. If this is important it should be given some context.
Answer to Query 5: The size of the molecule is not an important information and was therefore removed from the revised manuscript.
Query 6: There is a need to comment and add sections on
- long acting ART
- specifics of switching regimens for certain classes post-resistance development. Any clinical caveats to highlight?
- context of genotyping and impact on practice in low income settings
- mAb development and bnAb development
Answer to Query 6: The manuscript discusses lengthily all types ART and the implication of their setbacks on virologic failure and HIV/AIDS management. We therefore addressed the raised issues as follows:
- long acting ART was briefly discussed in the revised manuscript
- challenges of NGS genotyping in resource limited settings (RLS) were briefly highlighted in the revised manuscript
- mAb and bnAb drug candidates, advantages and disadvantages were discussed in the revised manuscript.
We could not find any clinical caveats to highlight specifics of switching regimens for certain classes post-resistance development.
Query 7: Authors and copy-editors should check for punctuation errors (e.g. line 659)
Answer to Query 7: Punctuation errors were addressed in the revised manuscript.

Reviewer 2 Report
In the current work, authors aimed to review current ART therapy, but most importantly their mechanism of action, and the mechanism of HIV resistance to these drugs. Moreover, they aimed to point out the implications of HIV resistance on the management of AIDS, and briefly discuss prospective ART drug candidates. This study is beneficial and detailed for the field of anti-HIV studies. Therefore, I recommend “accept for publication after minor revision”.
Here are the points:
-The enumeration as 1233 research papers: 546 in PubMed, 279 in Web of Science, 314 through Scopus, 94 through Science Direct does not look correct in Line 120. Because there is a big possibility for a paper to be found in both PubMed, Scopus or Science Direct.
-Authors should add a figure showing the chemical structures of molecules such as azidothymidine, tenofovir disoproxil fumarate (TDF), emtricitabine, efavirenz, etc.
-There is an underscore in the title of Figure 2.
-Figure 2 looks blurry and blue arrows should be rearranged.
- “on the central nervous system CNS” in Line 255, CNS should be in parenthesis.
-“Atripla” should be used with ® as it is a marketed name.
Minor editing of English language required.
Author Response
Responses to reviewer 2:
Query 1: The enumeration as 1233 research papers: 546 in PubMed, 279 in Web of Science, 314 through Scopus, 94 through Science Direct does not look correct in Line 120. Because there is a big possibility for a paper to be found in both PubMed, Scopus or Science Direct.
Answer to Query 1: In line 121 to 122, the sentence “Out of these, 1044 were excluded as they either contained redundant information or were irrelevant to the research topic” clearly addressed this issue.
Query 2: Authors should add a figure showing the chemical structures of molecules such as azidothymidine, tenofovir disoproxil fumarate (TDF), emtricitabine, efavirenz, etc.
Answer to Query 2: In our humble opinion, adding Chemical structures of all the mentioned drugs in the manuscript is a very cumbersome task and it will make the manuscript unnecessarily lengthy. However molecular structures of all the mentioned drugs are available online on drugbank website (https://go.drugbank.com/drugs/).
Query 3: There is an underscore in the title of Figure 2.
Answer to Query 3: This issue was resolved in the revised manuscript.
Query 4: Figure 2 looks blurry and blue arrows should be rearranged.
Answer to Query 4: A clearer figure was inserted in the revised manuscript.
Query 5: “on the central nervous system CNS” in Line 255, CNS should be in parenthesis.
Answer to Query 5: This issue was addressed in the revised manuscript.
Query 6: “Atripla” should be used with ® as it is a marketed name.
Answer to Query 6: ® was added to Atripla and other drugs marketed names.
